# Boosted CVaR Classification

**Runtian Zhai, Chen Dan, Arun Sai Suggala, Zico Kolter, Pradeep Ravikumar**
School of Computer Science
Carnegie Mellon University
Pittsburgh, PA, USA 15213
{rzhai,cdan,asuggala,zkolter,pradeepr}@cs.cmu.edu

## Abstract

Many modern machine learning tasks require models with high tail performance, *i.e.* high performance over the worst-off samples in the dataset. This problem has been widely studied in fields such as algorithmic fairness, class imbalance, and risk-sensitive decision making. A popular approach to maximize the model's tail performance is to minimize the CVaR (Conditional Value at Risk) loss, which computes the average risk over the tails of the loss. However, for classification tasks where models are evaluated by the $0/1$ loss, we show that if the classifiers are deterministic, then the minimizer of the average $0/1$ loss also minimizes the CVaR $0/1$ loss, suggesting that CVaR loss minimization is not helpful without additional assumptions. We circumvent this negative result by minimizing the CVaR loss over randomized classifiers, for which the minimizers of the average $0/1$ loss and the CVaR $0/1$ loss are no longer the same, so minimizing the latter can lead to better tail performance. To learn such randomized classifiers, we propose the Boosted CVaR Classification framework which is motivated by a direct relationship between CVaR and a classical boosting algorithm called LPBoost. Based on this framework, we design an algorithm called $\alpha$-AdaLPBoost. We empirically evaluate our proposed algorithm on four benchmark datasets and show that it achieves higher tail performance than deterministic model training methods.

## 1 Introduction

As machine learning continues to find broader usage, there is an increasing understanding of the importance of *tail performance* of models, in addition to their average performance. For instance, in datasets with highly imbalanced classes, the tail performance is the accuracy over the minority classes which have much fewer samples than the others. In the field of algorithmic fairness, where a dataset contains several demographic groups, the tail performance is the accuracy over certain underrepresented groups that normal machine learning models often neglect. In all these examples, it is crucial to design models with good tail performance that perform well across all parts/groups of the data domain, instead of just performing well on average.

Owing to its importance, a number of recent works have designed techniques to learn models with high tail performance [HSNL18, SKHL20, SRKL20]. Maximizing the tail performance is sometimes referred to as learning under *subpopulation shift*, in the sense that the testing distribution could consist of just a subpopulation of the training distribution. Most of the works on subpopulation shift fall into two categories. In the first, also referred to as the *domain-aware* setting, the dataset is divided into several predefined groups, and the goal is to maximize the *worst-group performance*, *i.e.* the minimum performance over all the groups. A number of methods, such as importance weighting [Shi00] and Group DRO [SKHL20, SRKL20], have been proposed for domain-aware subpopulation shift. However, the domain-aware setting is not always applicable, either because the groups can be hard to define, or because the group labels are not available. Thus, in the second category of

work on subpopulation shift, also referred to as the *domain-oblivious* setting, there are no pre-defined groups, and the goal is to maximize the model's performance over the worst-off samples in the dataset. Most previous work on domain-oblivious subpopulation shift [HSNL18, DN18, HNSS18, LBC+20, MHN21] measure the tail performance using the Distributionally Robust Optimization (DRO) loss, which is defined as the model's maximum loss over all distributions within a divergence ball around the training distribution. A popular instance is the $\alpha$-CVaR loss, defined as the model's average loss over the worst $\alpha \in (0, 1)$ fraction of the samples incurring the highest losses in the dataset.

Naturally one might think of maximizing a model's tail performance by directly minimizing the DRO loss or the $\alpha$-CVaR loss, which has been used by many previous work [HSNL18, DN18, XDKR20]. However, [HNSS18] proved the negative result that for classification tasks where models are evaluated by the zero-one loss, empirical risk minimization (ERM) achieves the lowest possible DRO loss given that the model is deterministic and the DRO loss is defined by some $f$-divergence function. We extend their result to the $\alpha$-CVaR loss which can be written as the limit of Rényi-divergence DRO losses (with a more direct and simpler proof due to the specialized case of CVaR). This is a very pessimistic result since it entails that there is no hope to get a better classifier than ERM so long as models are evaluated by a DRO (or CVaR) loss. So some previous work [HNSS18, LBC+20, MHN21] proposed to avoid this issue by making extra assumptions on the testing distribution (specifically, that the testing subpopulation can be represented by a parametric model), and changing the evaluation metric correspondingly to something other than a $f$-divergence DRO loss.

In this work, we take a different approach and show that no extra assumption is needed provided that we use *randomized models*. While the case of general DRO is more complicated, the reason why ERM achieves the lowest possible CVaR zero-one loss in the deterministic case is very simple: there exists a linear relationship between the CVaR zero-one loss and the average zero-one loss, so the former is non-decreasing with the latter. For randomized models, however, such a monotonic relationship no longer exists. Note that for any single test sample, the zero-one loss of the deterministic model is either 0 or 1, while the expected zero-one loss of the randomized model is a real number in $[0, 1]$, so that the randomized model can typically achieve lower $\alpha$-CVaR loss than the deterministic model. In fact, we can prove that if the two models have the same average accuracy, then the $\alpha$-CVaR zero-one loss of the randomized model is *consistently* lower than the deterministic one.

Motivated by the above analysis, we propose the framework of Boosted CVaR Classification to train ensemble models via Boosting. The key observation we make is that minimizing the $\alpha$-CVaR loss is equivalent to maximizing the objective of $\alpha$-LPBoost, a subpopulation-performance counterpart of a classical boosting variant known as LPBoost [DBST02]. Thus training with respect to the $\alpha$-CVaR loss can be related to a goal of *boosting an unfair learner*, which always produces a model with low average loss on any reweighting of the training set, to obtain a fair ensemble classifier with low $\alpha$-CVaR loss for a fixed $\alpha$. Note that this is in contrast to the classical boosting, which boosts a weak learner to produce an ensemble model with better average performance. We can thus show that $\alpha$-CVaR training is equivalent to a sequential min-max game between a Boosting algorithm and an unfair learner, in which the Boosting algorithm provides the sample weights and the unfair learner provides base models with low average loss with respect to these weights. After all base models are trained, we compute the optimal model weights. At inference time, we first randomly sample a base model according to the model weights, and then predict with the model. Thus, the final ensemble model is a linear combination of all the base models.

This paper is organized as follows: In Section 2, we provide the necessary background of subpopulation shift and CVaR, and show that ERM achieves the lowest CVaR zero-one loss in classification tasks with deterministic classifiers. In Section 3 we show how to boost an unfair learner: we first show that minimizing the CVaR loss is equivalent to maximizing the LPBoost objective in Section 3.1; based on this observation, we propose the Boosted CVaR Classification framework and implement an algorithm that uses LPBoost for CVaR classification in Section 3.2; Then, in order to improve computation efficiency, we implement another algorithm called $\alpha$-AdaLPBoost in Section 3.3. Finally, in Section 4 we empirically evaluate the proposed method on popular benchmark datasets.

## 1.1 Related Work

**Caveats of DRO.** DRO was first applied to domain-oblivious subpopulation shift tasks in [HSNL18], in which the authors proved that the DRO loss is an upper bound of the worst-case loss over $K$ groups (group CVaR) for fairness problems. [DN18] analyzed the convergence rate of

DRO for the Cressie-Read family of $f$-divergence. However, [HNSS18] showed that minimizing $f$-divergence DRO over deterministic function classes, with respect to the zero-one classification loss, yields the same minimizer as that of ERM, provided that the former has loss less than one. [SKHL20] further showed that for highly flexible classifiers (such as many modern neural models) that achieve zero error on each training sample, both empirical average risks and DRO risks are zero, so that the model is not specifically focusing on population DRO objective. [SRKL20] made the related point that the Group DRO objective with respect to the zero one loss is prone to overfit.

**Boosting.** Boosting is a classic algorithm in machine learning dating back to AdaBoost proposed in [Sch90]. See [Sch03] for a survey of the early works. Motivated by the success of AdaBoost, many later works proposed variants of Boosting, such as AdaBoost$_v^*$ [RWST05], AdaBoost$_{\ell_1}$ [SL09], LPBoost [DBST02], SoftBoost [RWG07] and entropy regularized LPBoost [WGV08]. There are some previous works that apply ensemble methods to tasks related to subpopulation shift: see [GFB+11] for a survey of Bagging, Boosting and hybrid methods for the class imbalance problem, while [IN19, BHL19] used AdaBoost to improve algorithmic fairness.

## 2  Preliminaries

In this section we provide the necessary background on subpopulation shift, DRO and CVaR, in the context of classification, which is the focus of the paper. Particularly, we will demonstrate that in classification tasks, ERM achieves the lowest CVaR loss, so there is no gain in using CVaR compared to ERM. Then, we show that using randomized models can circumvent this problem.

Denote the input space by $\mathcal{X}$, the label space by $\mathcal{Y}$, and the data domain by $\mathcal{Z} = \mathcal{X} \times \mathcal{Y}$. We are given a dataset $\{z_i = (\boldsymbol{x}_i, y_i)\}_{i=1}^n$ that *i.i.d.* sampled from some underlying data distribution. We assume that any input $\boldsymbol{x}$ has only one true label $y$, *i.e.* for any $\boldsymbol{x} \in \mathcal{X}$ there exists $y \in \mathcal{Y}$ such that $P(y \mid \boldsymbol{x}) = 1$. Given a family of classifiers $\mathcal{F}$, the goal of a subpopulation shift task is to train a classifier $F : \mathcal{X} \to \mathcal{Y} \in \mathcal{F}$ with high tail performance. Given a loss function $\ell : \mathcal{Y} \times \mathcal{Y} \to \mathbb{R}$, the standard training algorithm is empirical risk minimization (ERM) which minimizes the *empirical risk* defined as

$$\hat{\mathcal{R}}^\ell(F) = \frac{1}{n} \sum_{i=1}^n \ell(F(\boldsymbol{x}_i), y_i). \tag{1}$$

Denote the set of minimizers of the empirical risk by $F_{\text{ERM}^\ell}^* = \arg\min_{F \in \mathcal{F}} \hat{\mathcal{R}}^\ell(F)$. For classification tasks, the model performance is evaluated by the zero-one loss $\ell_{0/1}(\hat{y}, y) = \mathbf{1}_{\{\hat{y} \neq y\}}$. Although we can use different surrogate loss functions during training, at test time we always use the zero-one loss because we care about the accuracy, and the zero-one loss is equal to one minus the accuracy.

To quantitatively measure the tail performance of model $F$, we can use the $\alpha$-CVaR (Conditional Value at Risk) loss. For a fixed $\alpha \in (0, 1)$, the *$\alpha$-CVaR loss* is defined as

$$\text{CVaR}_\alpha^\ell(F) = \max_{\boldsymbol{w} \in \Delta_n, \boldsymbol{w} \preccurlyeq \frac{1}{\alpha n}} \sum_{i=1}^n w_i \ell(F(\boldsymbol{x}_i), y_i) \tag{2}$$

where $\Delta_n = \{(x_1, \cdots, x_n) : x_i \geq 0, x_1 + \cdots + x_n = 1\}$ is the unit simplex in $\mathbb{R}^n$. The $\alpha$-CVaR loss measures how well a model performs over the worst $\alpha$ fraction of the dataset. For instance, if $m = \alpha n$ is an integer, then the $\alpha$-CVaR loss is the average loss over the $m$ samples that incur the highest losses. We use *CVaR classification* to refer to classification tasks where models are evaluated by the CVaR loss. Denote the set of minimizers of the $\alpha$-CVaR loss by $F_{\text{CVaR}_\alpha^\ell}^* = \arg\min_{F \in \mathcal{F}} \text{CVaR}_\alpha^\ell(F)$.

The CVaR loss can be written as the limit of DRO (Distributional Robust Optimization) losses. For some divergence function $D$ between distributions, the DRO loss measures the model's performance over the worst-case distribution $Q \ll P$[1] within a ball w.r.t. divergence $D$ around the training distribution $P$. Formally, the DRO loss of model $F$ is defined as

$$\text{DRO}_{D,\rho}^\ell(F) = \sup_{Q \ll P} \{\mathbb{E}_Q[\ell(F(\boldsymbol{x}), y)] : D(Q \parallel P) \leq \rho\} \tag{3}$$

If we denote the Rényi-divergence by $D_\beta(P \parallel Q) = \frac{1}{\beta-1} \log \int (\frac{dP}{dQ})^\beta dQ$ , then the $\alpha$-CVAR loss is equal to the limit of $\text{DRO}_{D_\beta, -\log\alpha}^\ell$ as $\beta \to \infty$ (see Example 3 in [DN18]). Many previous works

---

[1] $Q$ is absolute continuous to $P$ (i.e. $Q \ll P$) if for any event $A$, $P(A) = 0 \Rightarrow Q(A) = 0$.

proposed to train a model with high tail performance by minimizing the CVaR loss or the DRO loss. However, the following result shows that for classification tasks, any model in $F^*_{\text{ERM}^{\ell_{0/1}}}$ is also the minimizer of the CVaR loss if $\mathcal{F}$ only contains deterministic models, i.e. every $F \in \mathcal{F}$ is a deterministic mapping $F : \mathcal{X} \mapsto \mathcal{Y}$ (all proofs can be found in Appendix A):

**Proposition 1.** *If $\mathcal{F}$ only contains deterministic models, then for any model $F \in \mathcal{F}$ and any $F^* \in F^*_{\text{ERM}^{\ell_{0/1}}}$, we have $\text{CVaR}_\alpha^{\ell_{0/1}}(F) \geq \text{CVaR}_\alpha^{\ell_{0/1}}(F^*)$. Moreover, if $\min_{F \in \mathcal{F}} \hat{\mathcal{R}}^{\ell_{0/1}}(F) < \alpha$, then we have $F^*_{\text{ERM}^{\ell_{0/1}}} = F^*_{\text{CVaR}_\alpha^{\ell_{0/1}}}$.*

[HNSS18] showed that a counterpart of the above result holds for any $f$-divergence DRO loss. As noted earlier, we can write the $\alpha$-CVaR loss as the limit of the Rényi family of $f$-divergence DRO losses. However our proof is much more direct and simple, and proceeds by showing the following simple monotonic relationship between the average zero-one loss and the $\alpha$-CVaR zero-one loss: $\text{CVaR}_\alpha^{\ell_{0/1}}(F) = \min\left\{1, \frac{1}{\alpha}\hat{\mathcal{R}}^{\ell_{0/1}}(F)\right\}$, so the $\alpha$-CVaR loss is non-decreasing with the ERM loss. This is a very pessimistic result, since it entails that there is no hope to obtain a better model than ERM no matter what learning algorithm we use for CVaR classification.

For the DRO context, some previous papers [HNSS18, LBC$^+$20, MHN21] propose to avoid this issue by making extra assumptions on the testing distribution, so as to change the evaluation metric to some function other than the $f$-divergence DRO loss. In this work, however, we take a completely different approach: we show that the above difficulty can be circumvented without any extra assumptions by using randomized models[2]. For a randomized model $F$, its empirical risk and $\alpha$-CVaR loss is defined as the expectation of (1) and (2), where the expectation is taken over the randomness of $F$. For randomized models, the monotonic relationship in Proposition 1 does not exist, and they can achieve lower $\alpha$-CVaR zero-one loss than deterministic models. For example, if we have a deterministic model with average accuracy 90%, then the 10%-CVaR zero-one loss of this model is 1. However, if we have 5 deterministic models with average accuracy 90%, such that each sample is classified correctly by at least 4 of the 5 models, then the 10%-CVaR zero-one loss of the average of the 5 models is only 0.2, though its average accuracy is still 90%. Furthermore, we can prove that:

**Proposition 2.** *Let $F$ be a deterministic model, and $F'$ be any randomized model whose average zero-one loss is the same as that of $F$. Then, for any $\alpha \in (0, 1)$, $\text{CVaR}_\alpha^{\ell_{0/1}}(F') \leq \text{CVaR}_\alpha^{\ell_{0/1}}(F)$.*

This result implies that the tail performance of a randomized model is consistently higher than a deterministic model with the same average performance. In a nutshell, we have proved that for CVaR classification, ERM is the best deterministic model learning algorithm, and randomized models can achieve higher performance than deterministic models.

## 3 Boosted CVaR Classification

In this section, we propose the framework of Boosted CVaR Classification, which learns ensemble models with high tail performance via Boosting. An ensemble model consists of $T$ base models $f^1, \cdots, f^T : \mathcal{X} \to \mathcal{Y}$ and a distribution $\boldsymbol{\lambda} = (\lambda^1, \cdots, \lambda^T) \in \Delta_T$ over the models. $\boldsymbol{\lambda}$ is called the model weight vector. At inference time, we first sample a model $f^t$ according to $\boldsymbol{\lambda}$, and then predict with the model. Denote the zero-one loss of base model $f^t$ over sample $z_i$ by $\ell_i^t$. Then, the $\alpha$-CVaR zero-one loss of the ensemble model $F = (f^1, \cdots, f^T, \boldsymbol{\lambda})$ is[3]

$$\text{CVaR}_\alpha^{\ell_{0/1}}(F) = \text{CVaR}_\alpha^{\ell_{0/1}}(f^1, \cdots, f^T, \boldsymbol{\lambda}) = \max_{\boldsymbol{w} \in \Delta_n, \boldsymbol{w} \preccurlyeq \frac{1}{\alpha n}} \sum_{i=1}^n w_i \sum_{t=1}^T \lambda_t \ell_i^t \tag{4}$$

The motivation of this framework comes from a direct relationship between the CVaR loss and the objective of a variant of Boosting we call $\alpha$-LPBoost, which shows that training with respect to the $\alpha$-CVaR loss can be related to the goal of *boosting an unfair learner* (Section 3.1). Thus in Section 3.2, we present the Boosted CVaR Classification framework which formulates the training

---

[2]If $F$ is a randomized model, then for any $x$, $F(x)$ is a random variable over $\mathcal{Y}$, i.e. for any $x, y$, $P(F(x) = y)$ is a real number in $[0, 1]$ instead of binary.

[3]The notion $F = (f^1, \cdots, f^T, \boldsymbol{\lambda})$ means that the ensemble model $F$ consists of base models $f^1, \cdots, f^T$ and model weight vector $\boldsymbol{\lambda}$.

process as a sequential game between the training algorithm and the unfair learner, and implement an algorithm that uses (Regularized) LPBoost for CVaR classification. Finally, in Section 3.3 we implement another algorithm which we name $\alpha$-AdaLPBoost that is computationally more efficient.

## 3.1 $\alpha$-LPBoost

Suppose we have already obtained $t$ base models $\{f^s\}_{s \in [t]}$, and the $s$-th function $f^s$ incurs losses $\{\ell_i^s\}_{i \in [n]}$ on the $n$ samples $\{z_i\}_{i \in [n]}$. Consider the following primal/dual linear programs:

**Dual:**                                                    **Primal:**

$$\min_{\boldsymbol{w}, \gamma} \quad \gamma$$
$$\text{s.t.} \quad \langle \boldsymbol{w}, \boldsymbol{\ell}^s \rangle \geq 1 - \gamma; \; \forall s \in [t] \quad (5)$$
$$\boldsymbol{w} \in \Delta_n, \boldsymbol{w} \preccurlyeq \frac{1}{\alpha n}$$

$$\max_{\boldsymbol{\lambda}, \rho} \quad \rho - \frac{1}{\alpha n} \sum_{i=1}^{n} (\rho - 1 + \sum_{s=1}^{t} \lambda_s \ell_i^s)_+ \quad (6)$$
$$\text{s.t.} \quad \boldsymbol{\lambda} \in \Delta_t$$

where $(x)_+ = \max\{x, 0\}$. Note that the primal problem can be written as a linear program by introducing slack variables $\psi_i = (\rho - 1 + \sum_{s=1}^{t} \lambda_s \ell_i^s)_+$. See the full derivation of this primal-dual linear program in Appendix A.3. Let us denote the optimal dual objective by $\gamma_*^t$ and the optimal primal objective by $\rho_*^t$. For this linear program, strong duality holds, i.e. $\rho_*^t = \gamma_*^t$. We refer to these primal/dual programs as $\alpha$-LPBoost, since it can be seen as subpopulation-performance counterpart of a classical variant of Boosting called LPBoost [DBST02].

Intuitively, the dual problem is trying to find a $\boldsymbol{w}$ such that every $f^s$ has a high weighted average loss $\langle \boldsymbol{w}, \boldsymbol{\ell}^s \rangle$ with respect to $\boldsymbol{w}$. One might wonder what the primal problem is doing. The magical thing is that the primal problem is in fact *searching for the model weight vector $\boldsymbol{\lambda}$ that minimizes the $\alpha$-CVaR zero-one loss of the ensemble model consisting of $f^1, \cdots, f^t$*, as shown by the following proposition:

**Proposition 3.** *For any $f^1, \cdots, f^t$, we have the following relationship between the $\alpha$-LPBoost objective and the $\alpha$-CVaR zero-one loss:*

$$\rho_*^t = \gamma_*^t = 1 - \min_{\boldsymbol{\lambda} \in \Delta_t} \text{CVaR}_\alpha^{\ell_{0/1}}(f^1, \cdots, f^t, \boldsymbol{\lambda}) \quad (7)$$

*and the optimal solution of the primal problem $\boldsymbol{\lambda}^*$ achieves the minimum in (7).*

This result also shows a direct relationship between CVaR and LPBoost: minimizing the $\alpha$-CVaR loss is equivalent to maximizing the $\alpha$-LPBoost objective. Thus, the problem now becomes how to maximize $\gamma_*^t$. Note that we can rewrite the first constraint of the dual problem as $\gamma \geq 1 - \langle \boldsymbol{w}, \boldsymbol{\ell}^s \rangle$ for all $s$, so $\gamma$ is the accuracy of the best $f^s$. Therefore, we can increase $\gamma$ by training a new model $f^{t+1}$ whose weighted average loss with respect to $\boldsymbol{w}$, i.e. $\langle \boldsymbol{w}, \boldsymbol{\ell}^{t+1} \rangle$, is small. We can repeat this process until we have obtained sufficient base models so that there is no such $\boldsymbol{w}$ that makes $\gamma_*^t$ small. Then, we can obtain the optimal model weight vector $\boldsymbol{\lambda}^*$ by solving the primal problem of $\alpha$-LPBoost.

## 3.2 The Boosted CVaR Classification Framework

Motivated by the above analysis, we design the Boosted CVaR Classification framework that formulates the training process outlined in the previous section as a sequential game between a boosting algorithm and an *unfair learner* $\mathcal{L}$. We make the following assumption on $\mathcal{L}$:

**Assumption 1.** *We have access to an unfair learner $\mathcal{L}$ that takes any sample weight vector $\boldsymbol{w} \in \Delta_n$ as input, and always outputs a base model whose weighted average zero-one loss with respect to $\boldsymbol{w}$ is at most $g \in (0, 1)$. $g$ is called the* guarantee *of the learner $\mathcal{L}$.*

In each round, the boosting algorithm picks a sample weight vector $\boldsymbol{w}^t = (w_1^t, \cdots, w_n^t)$ and feeds it to the learner $\mathcal{L}$ which outputs a base model $f^t$ whose average loss w.r.t. $\boldsymbol{w}^t$ is at most $g$. The boosting algorithm goes as the following:

- For $t = 1, \cdots, T$,
    - Pick a sample weight vector $\boldsymbol{w}^t = (w_1^t, \cdots, w_n^t) \in \Delta_n$ and feed it to $\mathcal{L}$
    - Receive a base model $f^t$ from $\mathcal{L}$ such that $\sum_{i=1}^{n} w_i^t \ell_i^t \leq g$
- At the end of training, pick a model weight vector $\boldsymbol{\lambda} \in \Delta_T$ and return $F = (f^1, \cdots, f^T, \boldsymbol{\lambda})$

---

**Algorithm 1** (Regularized) $\alpha$-LPBoost for CVaR Classification

---

**Input:** Density of the test subpopulation $\alpha$, regularization coefficient $\beta$, number of base models $T$

 1: Initialization: $\boldsymbol{w}^1 = (\frac{1}{n}, \cdots, \frac{1}{n})$
 2: **for** $t = 1, \cdots, T$ **do**
 3:     Run learner $\mathcal{L}$ with weight vector $\boldsymbol{w}^t$
 4:     $\mathcal{L}$ outputs model $f^t$ with sample losses $\boldsymbol{\ell}^t = (\ell_1^t, \cdots, \ell_n^t)$
 5:     Solve the dual $\alpha$-LPBoost problem (5) or its regularized version (8) over the training set to get the sample weights $\boldsymbol{w}^{t+1}$
 6: Solve the primal $\alpha$-LPBoost problem (6) *over the validation set* to get the optimal $\boldsymbol{\lambda}$
 7: **return** $F = (f^1, \cdots, f^T, \boldsymbol{\lambda})$

---

To study the worst-case performance of the boosting algorithm, we can view $\mathcal{L}$ as an adversary: The boosting algorithm picks $\boldsymbol{w}^t$ and $\boldsymbol{\lambda}$ in order to minimize (4), while $\mathcal{L}$ picks $\boldsymbol{\ell}^t = (\ell_1^t, \cdots, \ell_n^t)$ under the constraint $\langle \boldsymbol{w}^t, \boldsymbol{\ell}^t \rangle \leq g$ in order to maximize (4).

Based on this framework, we implement Algorithm 1, which uses LPBoost to pick sample weights and model weights. For solving linear programs, there are a number of convex optimization solvers available. Now we prove a convergence rate theorem for this algorithm. To show that Algorithm 1 converges, we need to prove that with a sufficiently large $T$, the worst-case $\alpha$-CVaR loss of the ensemble model can be as close to $g$ as we want[4]. Ideally we would like $T$ to be in the order of $\log \frac{1}{\alpha}$. However, [RWG07] presented a counterexample in its Theorem 1 where $\alpha$-LPBoost requires $T = \Omega(\frac{1}{\alpha})$ base models to converge. To make $T$ logarithmic in $\frac{1}{\alpha}$, [WGV08] proposed (Entropy) Regularized $\alpha$-LPBoost, which adds regularization to the dual problem. At each iteration it solves the following convex problem for some regularization coefficient $\beta > 0$ to pick $\boldsymbol{w}$:

$$
\begin{aligned}
\min_{\boldsymbol{w}} \quad & \gamma - \frac{1}{\beta} H(\boldsymbol{w}) \\
\text{s.t.} \quad & \langle \boldsymbol{w}, \boldsymbol{\ell}^s \rangle \leq 1 - \gamma \ (s \in [t]), \ \boldsymbol{w} \in \Delta_n, \ \boldsymbol{w} \preccurlyeq \frac{1}{\alpha n}
\end{aligned}
\tag{8}
$$

where $H(\boldsymbol{w}) = -\sum_{i=1}^n w_i \log w_i$ is the entropy function. With the regularization, we can prove the following theorem:

**Theorem 4** (Theorem 1 in [WGV08]). *For any $\delta > 0$, if we run Regularized $\alpha$-LPBoost with $\beta = \max(\frac{2}{\delta} \log \frac{1}{\alpha}, \frac{1}{2})$, then we have* $\mathrm{CVaR}_\alpha^{\ell_{0/1}}(F) \leq g + \delta$ *with $T$ base models where*

$$
T = \max\left\{ \frac{32}{\delta^2} \log \frac{1}{\alpha}, \frac{8}{\delta} \right\}
\tag{9}
$$

**Connection to AdaBoost.** [SL09] proved that classical Regularized LPBoost (i.e. $\alpha$-LPBoost with $\alpha = 1/n$) is equivalent to AdaBoost[FS97] with $\ell_1$ regularization (see its Section 3.2).

### 3.3 $\alpha$-AdaLPBoost

We have proved that the $\alpha$-CVaR loss achieved by Regularized $\alpha$-LPBoost can be arbitrarily close to $g$ with $T = O(\log \frac{1}{\alpha})$. However, one disadvantage of Algorithm 1 is that it needs to train a different set of $T$ base models for each $\alpha$. On the other hand, since $\alpha$ controls the trade-off between fairness and average performance, in practice we might want to change $\alpha$ from time to time. For example, we might want to tune down $\alpha$ to make the model fairer. Thus, it would be more efficient if we could train only one set of $T$ base models for all $\alpha$, and by adjusting the model weight vector $\boldsymbol{\lambda}$ for different $\alpha$ we could still achieve tail performance comparable to Regularized $\alpha$-LPBoost.

To this end, we first use the classical Regularized LPBoost (with $\alpha = 1/n$) to pick $\boldsymbol{w}^t$ to train the base models, since the resulting base models would also be suitable for more general $\alpha > 1/n$. Following [SL09], we solve this step using a variant of AdaBoost. We then pick the model weights by solving the $\alpha$-LPBoost primal problem after all base models are trained. The method, which we

---

[4]The $\alpha$-CVaR loss is lower bounded by $g$, because $\mathcal{L}$ can always output a model whose average loss over the uniform distribution of $z_1, \cdots, z_n$ is at least $g$, so that the average loss is at least g.

---

**Algorithm 2** $\alpha$-AdaLPBoost

---

**Input:** Step size $\eta$, density of the test subpopulation $\alpha$, number of base models $T$
 1: Initialization: $\boldsymbol{w}^1 = (\frac{1}{n}, \cdots, \frac{1}{n})$
 2: **for** $t = 1, \cdots, T$ **do**
 3:     Run learner $\mathcal{L}$ with weight vector $\boldsymbol{w}^t$
 4:     $\mathcal{L}$ outputs model $f^t$ with sample losses $\boldsymbol{\ell}^t = (\ell_1^t, \cdots, \ell_n^t)$
 5:     Pick $\boldsymbol{w}^{t+1} \in \Delta_n$ such that $w_i^{t+1} \propto \exp(\eta \sum_{s=1}^t \ell_i^s)$
 6: Solve the $\alpha$-LPBoost primal problem (6) *over the validation set* to get the optimal $\boldsymbol{\lambda}$
 7: **return** $f^1, \cdots, f^T$ and $\boldsymbol{\lambda}$

---

denote by $\alpha$-AdaLPBoost, is listed in Algorithm 2. The difference between Algorithm 2 and the original AdaBoost is that AdaBoost picks $w_i^{t+1} \propto w_i^t \beta_t^{\ell_i^t}$ where $\beta_t = \frac{\epsilon_t}{1-\epsilon_t}$ and $\epsilon_t = \sum_{i=1}^n w_i^t \ell_i^t$ is the weighted average loss, whereas Algorithm 2 picks $w_i^{t+1} \propto w_i^t \exp(\eta \ell_i^t)$ for a constant $\eta$, which we find achieves better performance than AdaBoost in our experiments.

The advantage of using our two-step approach is that when $\alpha$ changes, we only need to adjust the model weight vector $\boldsymbol{\lambda}$ without training an entirely new set of base models. We next show that the two-step approach is not just intuitively reasonable, but comes with strong theoretical guarantees. To begin with, we consider a mixed algorithm called "AdaBoost + Average", where we train the based models with AdaBoost (as in Algorithm 2) and output the average of the base models (such that $\boldsymbol{\lambda} = (\frac{1}{T}, \cdots, \frac{1}{T})$). For AdaBoost + Average, we have the following result:

**Theorem 5.** *For any $\delta > 0$, and for $T = O(\frac{\log n}{\delta^2})$, the training $\alpha$-CVaR zero-one loss of the ensemble model given by AdaBoost + Average is at most $g + \delta$ if we set $\eta = \sqrt{8 \log n / T}$.*

The theorem guarantees that AdaBoost + Average can achieve low $\alpha$-CVaR zero-one loss. Next, note that the $\alpha$-CVaR zero-one loss of $\alpha$-AdaLPBoost is upper bounded by the minimum of those of ERM and AdaBoost + Average, because $\alpha$-LPBoost ensures that the $\boldsymbol{\lambda}$ it picks achieves the lowest $\alpha$-CVaR zero-one loss, while ERM corresponds to $\boldsymbol{\lambda} = (1, 0, \cdots, 0)$ and AdaBoost + Average corresponds to $\boldsymbol{\lambda} = (\frac{1}{T}, \frac{1}{T}, \cdots, \frac{1}{T})$. In other words, the theorem ensures that the tail performance of AdaBoost + Average is high, and $\alpha$-AdaLPBoost achieves a tail performance no lower than that.

### 3.4 Discussion

**Generalization Bound for the CVaR Loss.** In our analysis, we only provide theoretical guarantees on the training $\alpha$-CVaR zero-one loss for the methods we propose. One might be curious about the generalization capability of boosting with respect to the CVaR loss. A recent work [KLPR20] proved a generalization bound for the CVaR loss under the assumption that the hypothesis set $\mathcal{F}$ has a finite VC-dimension, and it is well-known that if $\mathcal{F}$ has a finite VC-dimension, then the set of ensemble models based upon $\mathcal{F}$ also has a finite VC-dimension (see e.g. Section 6.3.1 in [MRT18]). However, in the VC-dimension-based analysis, the generalization error increases with $T$, while in practice it usually decreases with $T$. Thus, people use the margin-based analysis to obtain a tighter bound. We leave the margin-based analysis of Boosted CVaR Classification to future work.

**Sensitivity to Outliers.** An algorithm that maximizes the tail performance of a model can be very sensitive to outliers. This is because such an algorithm puts more weight on samples on which the model has high losses, while intuitively outliers tend to incur high losses. Consequently, the algorithm puts more weight on outliers. A recent work [ZDKR21] proposed to solve this problem with an algorithm called DORO, which ignores a fraction of the samples with the highest losses within the minibatch for each iteration. DORO can also be combined with our Boosting algorithms.

**Connection to (Domain-Aware) Group DRO.** Our algorithm has a strong connection to Group DRO proposed in [SKHL20] for domain-aware subpopulation tasks. Group DRO assumes that the dataset is divided into $K$ groups and minimizes the model's maximum loss over the $K$ groups with AdaBoost. We can obtain Group DRO from AdaBoost + Average by choosing $n = K$, $m = 1$ and defining $\ell_i^t$ as the loss of model $f^t$ over group $i$. Our algorithm suggests that we can improve Group DRO if we choose the model weights with LPBoost.

## 4 Experiments

### 4.1 Setup

**Datasets.** We conduct our experiments on four datasets: COMPAS [LMKA16], CelebA [LLWT15], CIFAR-10 and CIFAR-100 [KH+09]. The first, COMPAS, is a tabular recidivism dataset widely used in fairness research. The second, CelebA, with the target label as whether a person's hair is blond or not, was used in [SKHL20] to evaluate their proposed Group DRO algorithm. And finally, CIFAR-10 and CIFAR-100 are two widely used image datasets. Both COMPAS and CelebA are binary classification tasks, while CIFAR-10 is 10-class and CIFAR-100 is 100-class classification. On COMPAS we use the training set as the validation set because the dataset is very small. CelebA has its official train-validation split. On CIFAR-10 and CIFAR-100 we take out 10% of the training samples and use them for validation. Our selection of datasets covers different types, complexity, and both binary and multi-class classification.

**Learner $\mathcal{L}$.** We implement the unfair learner $\mathcal{L}$ as follows: given a sample weight vector $\boldsymbol{w} = (w_1, \cdots, w_n) \in \Delta_n$, for each iteration we sample a minibatch with replacement according to the probability distribution $\boldsymbol{w}$, and then update the model parameters with ERM over the minibatch (by minimizing the cross-entropy loss). The learner stops and returns the model after a fixed number of iterations, with the learning rate decayed twice during training. See our code for training details.

**Training.** We use a three-layer feed-forward neural network with ReLU activations on COMPAS, a ResNet-18 [HZRS16] on CelebA, a WRN-28-1 [ZK16] on CIFAR-10 and a WRN-28-10 on CIFAR-100. On each dataset, we first warmup the model with a few epochs of ERM, and then train $T = 100$ base models on COMPAS and CIFAR-10, and $T = 50$ base models on CelebA and CIFAR-100 from the warmup model with the sample weights given by the boosting algorithms. We train our models with CPU on COMPAS and with one NVIDIA GTX 1080ti GPU on other datasets. We solve linear programs with the CVXPY package [DB16, AVDB18], which at its core invokes MOSEK [ApS21] and ECOS [DCB13] for optimization.[5]

On each dataset, we run ERM and $\alpha$-AdaLPBoost with different values of $\alpha$. Here ERM refers to the deterministic version, in which a single base model is trained and used as the final model. For $\alpha$-AdaLPBoost, we choose $\eta = 1.0$ on all datasets which is close to the theoretical optimal value $\eta = \sqrt{8 \log n / T}$. We also compare $\alpha$-AdaLPBoost with AdaBoost + Average in order to demonstrate the effectiveness of selecting $\boldsymbol{\lambda}$ with LPBoost. To simultaneously compare the performances under different $\alpha$, we plot the $\alpha$-CVaR zero-one loss vs $\alpha$ curve for each method on every dataset.

### 4.2 Results

We plot the experimental results in Figure 1. Each experiment is repeated five times with different random seeds, and we plot the 95% confidence interval for each experiment. From the figure, we can see that the results are very consistent across different random seeds. It is also remarkable that on every dataset, the $\alpha$-CVaR loss curve of $\alpha$-AdaLPBoost almost overlaps with the minimum of ERM and AdaBoost + Average. From the plots we can make the following observations:

- When $\alpha$ is large, the $\alpha$-CVaR loss is close to the average loss, and we can see that the $\alpha$-CVaR loss of $\alpha$-AdaLPBoost is very close to that of ERM. We also plot the test average loss of the three methods on CIFAR-10 and CIFAR-100 in Figure 2, from which we can see that there is a gap between the ERM line and AdaBoost + Average line, and the average loss curve of $\alpha$-AdaLPBoost mostly lies between them.

- When $\alpha$ is small, the $\alpha$-CVaR loss of $\alpha$-AdaLPBoost is close to that of AdaBoost + Average, which is much lower than that of ERM. Recall our initial theoretical results on the equivalence of minimizers of the average loss and the CVaR loss for deterministic models, ERM achieves the lowest $\alpha$-CVaR loss among all deterministic models, so the results demonstrate that the ensemble model achieves higher tail performance than any deterministic model.

Overall, we find that $\alpha$-AdaLPBoost can automatically choose between high average performance (when $\alpha$ is big) and high tail performance (when $\alpha$ is small).

---

[5]Please refer to their official websites for license information.

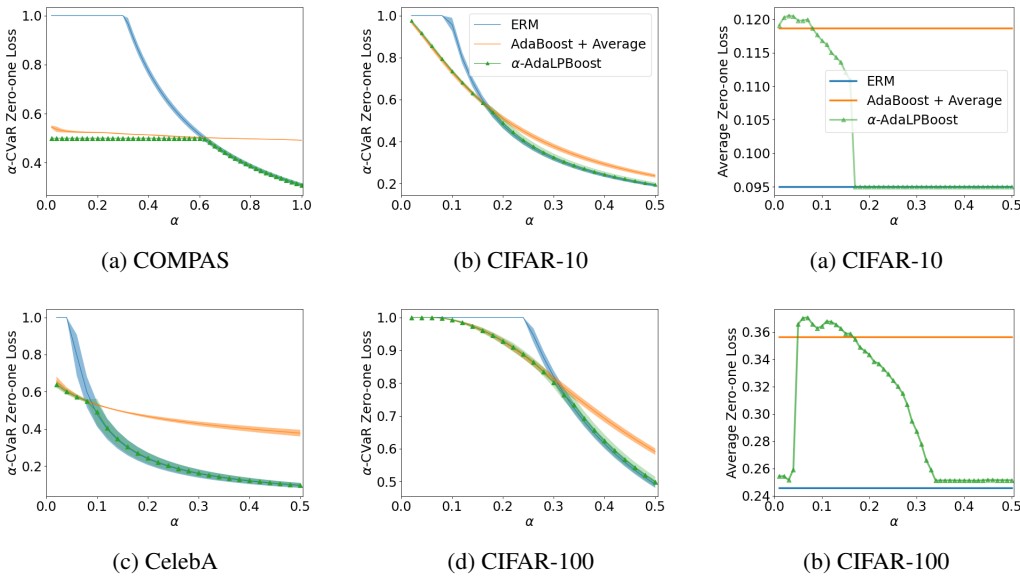

Figure 1: Test $\alpha$-CVaR zero-one loss.

Figure 2: Test average loss.

One might ask why $\alpha$-AdaLPBoost does not achieve lower average loss than ERM in our experiments, given that the model class of the ensemble models is larger than that of the deterministic models. The reason is that the model class we use for deterministic models (e.g. ResNet) is already very complex, so the average performance of ERM is high enough, and its loss mainly comes from the generalization gap instead of insufficient capacity of representation. In fact, we find that when $\alpha$ is big, the model weight vector produced by LPBoost is very close to $(1, 0, 0, \cdots, 0)$, i.e. almost all the weight is put on the first ERM model. That is why the $\alpha$-AdaLPBoost curve and the ERM curve overlap when $\alpha$ is big.

**Comparison with Regularized LPBoost.** Now we empirically show that $\alpha$-AdaLPBoost can achieve performance comparable to Regularized LPBoost. In Figure 3 we plot the $\alpha$-CVaR loss of ERM, $\alpha$-AdaLPBoost and Regularized LPBoost ($\beta = 100$) on CIFAR-10. The plot clearly shows that there is almost no gap between the performance of $\alpha$-AdaLPBoost and Regularized LPBoost, so we can safely replace the latter with the former for computational efficiency.

**Convergence.** Finally, we empirically examine the convergence rate of $\alpha$-AdaLPBoost by studying how the $\alpha$-CVaR loss changes with $T$. In Figure 4 we plot the results on CIFAR-10 and CIFAR-100, which show that AdaLPBoost converges slowly after $T = 30$. Theoretically, for a dataset with $n = 50000$ samples, in order to acheive $\delta < 0.1$ in Theorem 5, we need $T$ to be at least 500, which would take a huge amount of time. Note that $T = 30$ does not mean that the training time is 30 times more, because we can train each base model for fewer iterations since it is initialized from a warmup model and only needs finetune. In our experiments, the training time of AdaBoost is 3-6 times the training time of ERM if $T = 30$.

## 5   Conclusion

In this work, we addressed an issue raised by previous work that no deterministic model learning algorithm could be better than ERM for DRO classification (which we show formally also extends to CVaR) by learning randomized models. Specifically we proposed the Boosted CVaR Classification framework which is motivated by the direct relationship between $\alpha$-CVaR and $\alpha$-LPBoost, which is a sub-population variant of the classical LPBoost algorithm for classification. To further improve the computational efficiency, we implemented the $\alpha$-AdaLPBoost algorithm. In our experiments, we showed that the ensemble models produced by $\alpha$-AdaLPBoost achieved higher tail performance

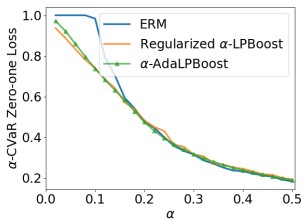 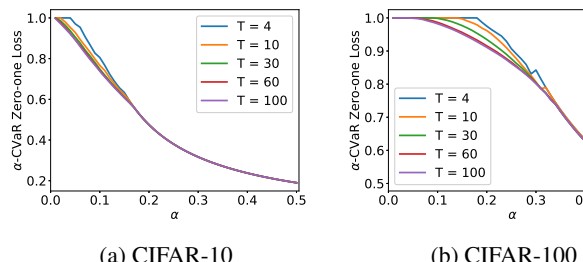

(a) CIFAR-10          (b) CIFAR-100

Figure 3: Comparing between $\alpha$-AdaLPBoost and Regularized $\alpha$-LPBoost on CIFAR-10.

Figure 4: Test $\alpha$-CVaR zero-one loss of $\alpha$-AdaLPBoost under different values of $T$.

than deterministic models, and the algorithm could automatically choose between high average performance and high tail performance under different values of $\alpha$.

One caveat of using a randomized model is that one might be able to game the model via repeatedly using it. For example, when applying for a credit card, one can submit the application repeatedly until it gets approved if the decision is given by a randomized model. It is important to study how to improve the tail performance in this scenario where the model can be used repeatedly, which we leave as an open question.

One future direction is the application of ensemble methods to the fairness without demographics problem, in which the dataset is divided into several groups which are unknown during training, and the goal is to minimize the model's worst-group loss, i.e. its maximum loss over all the groups. The worst-group loss can be upper bounded by the CVaR loss or certain families of DRO loss, but the bound is loose, and the model with the lowest CVaR loss is not guaranteed to achieve the lowest worst-group loss. Due to the difficulty of the problem, some recent works considered the scenario where a small set of samples with group labels is provided after training and before testing. Ensemble methods can be useful in this scenario: we can train $T$ base models, and then solve a linear program to obtain the optimal model weight vector with the provided validation set with no need of training new models. We leave the design of such algorithms to future work.

**Social Impact.**  Subpopulation shift has been widely studied to improve the fairness of machine learning, which is of great social importance. Models with high tail performance are considered fair because they perform well on all parts of the data domain. In this work we show how to improve the tail performance by learning ensemble models, which is a great contribution to the area of fair machine learning. However, we also observe that ensemble methods improve the tail performance by lowering the accuracy over samples in the majority group. Such trade-offs are nonetheless inevitable under the assumption that the average accuracy does not increase. It is an interesting sociological question to what extent it is just to improve fairness by sacrificing the majority group.

**Code.**  Codes for this paper can be found at: `https://github.com/RuntianZ/boosted_cvar`.

## Acknowledgments and Disclosure of Funding

This research is supported by DARPA Guaranteeing AI Robustness against Deception (GARD) under the cooperative agreement number HR00112020006 and the official title "Provably Robust Deep Learning".

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
