# Boosted CVaR Classification
## (Supplementary Material)

**Runtian Zhai, Chen Dan, Arun Sai Suggala, Zico Kolter, Pradeep Ravikumar**
School of Computer Science
Carnegie Mellon University
Pittsburgh, PA 15213
{rzhai,cdan,asuggala,zkolter,pradeepr}@cs.cmu.edu

## A  Proofs

### A.1  Proof of Proposition 1

We have the following relationship: $\text{CVaR}_\alpha^{\ell_{0/1}}(F) = \max_{\boldsymbol{w}\in\Delta_n, \boldsymbol{w}\preccurlyeq\frac{1}{\alpha n}} \sum_{i=1}^n w_i \mathbf{1}_{\{F(\boldsymbol{x}_i)\neq y_i\}} = \min\{1, \frac{1}{\alpha n}\sum_{i=1}^n \mathbf{1}_{\{F(\boldsymbol{x}_i)\neq y_i\}}\} = \min\{1, \frac{1}{\alpha}\text{ERM}^{\ell_{0/1}}(F)\}$. Thus, $\text{CVaR}_\alpha^{\ell_{0/1}}(F) \geq \text{CVaR}_\alpha^{\ell_{0/1}}(F^*)$ because $\text{ERM}^{\ell_{0/1}}(F) \geq \text{ERM}^{\ell_{0/1}}(F^*)$, so $F^*_{\text{ERM}^{\ell_{0/1}}} \subset F^*_{\text{CVaR}_\alpha^{\ell_{0/1}}}$.

If $\min_F \text{ERM}^{\ell_{0/1}}(F) < \alpha$, then for all $F$, we have$\text{CVaR}_\alpha^{\ell_{0/1}}(F) = \frac{1}{\alpha}\text{ERM}^{\ell_{0/1}}(F)$. Thus, $F^*_{\text{ERM}^{\ell_{0/1}}} = F^*_{\text{CVaR}_\alpha^{\ell_{0/1}}}$. $\square$

### A.2  Proof of Proposition 2

For a deterministic model $F$, we have $\text{CVaR}_\alpha^{\ell_{0/1}}(F) = \min\{1, \frac{1}{\alpha}\text{ERM}^{\ell_{0/1}}(F)\}$. For a randomized model $F'$ such that $\text{ERM}^{\ell_{0/1}}(F') = \text{ERM}^{\ell_{0/1}}(F)$, we have $\text{CVaR}_\alpha^{\ell_{0/1}}(F') \leq 1$ and

$$
\begin{aligned}
\text{CVaR}_\alpha^{\ell_{0/1}}(F') &= \max_{\boldsymbol{w}\in\Delta_n, \boldsymbol{w}\preccurlyeq\frac{1}{\alpha n}} \sum_{i=1}^n w_i P(F'(\boldsymbol{x}_i)\neq y_i) \leq \sum_{i=1}^n \frac{1}{\alpha n}P(F'(\boldsymbol{x}_i)\neq y_i)\\
&= \frac{1}{\alpha}\text{ERM}^{\ell_{0/1}}(F') = \frac{1}{\alpha}\text{ERM}^{\ell_{0/1}}(F)
\end{aligned}
\tag{10}
$$

Thus, $\text{CVaR}_\alpha^{\ell_{0/1}}(F') \leq \text{CVaR}_\alpha^{\ell_{0/1}}(F)$. $\square$

### A.3  Derivation of the Primal-Dual Formulation of $\alpha$-LPBoost

The primal problem of $\alpha$-LPBoost is

$$
\begin{aligned}
\max_{\boldsymbol{\lambda},\rho} \quad & \rho - \frac{1}{\alpha n}\sum_{i=1}^n (\rho - 1 + \sum_{s=1}^t \lambda^s \ell_i^s)_+\\
\text{s.t.} \quad & \boldsymbol{\lambda} \in \Delta_t
\end{aligned}
\tag{11}
$$

35th Conference on Neural Information Processing Systems (NeurIPS 2021), Sydney, Australia.

Introducing slack variables $\psi_i = (\rho - 1 + \sum_{s=1}^{t} \lambda^s \ell_i^s)_+$, the primal problem can be written as a linear program:

$$\max_{\boldsymbol{\lambda}, \rho, \boldsymbol{\psi}} \quad \rho - \frac{1}{\alpha n} \sum_{i=1}^{n} \psi_i$$
$$\text{s.t.} \quad \boldsymbol{\lambda} \in \Delta_t \tag{12}$$
$$\psi_i \geq 0, \ \psi_i \geq \rho - 1 + \sum_{s=1}^{t} \lambda^s \ell_i^s, \ \forall i \in [n]$$

The Lagrangian of this problem is

$$
\begin{aligned}
\mathcal{L}(\boldsymbol{\lambda}, \rho, \boldsymbol{\psi}, \boldsymbol{w}, \boldsymbol{\mu}, \boldsymbol{\nu}, \beta) =& -\rho + \frac{1}{\alpha n} \sum_{i=1}^{n} \psi_i - \sum_{s=1}^{t} \mu_s \lambda_s + \beta(\sum_{s=1}^{t} \lambda_s - 1) \\
& - \sum_{i=1}^{n} \nu_i \psi_i - \sum_{i=1}^{n} w_i (\psi_i - \rho + 1 - \sum_{s=1}^{t} \lambda_s \ell_i^s) \\
=& (\sum_{i=1}^{n} w_i - 1)\rho + \sum_{i=1}^{n} (\frac{1}{\alpha n} - \nu_i - w_i)\psi_i \\
& + \sum_{s=1}^{t} (\beta - \mu_s + \sum_{i=1}^{n} w_i \ell_i^s)\lambda_s - \beta - \sum_{i=1}^{n} w_i
\end{aligned}
\tag{13}
$$

The dual problem is $\max_{\boldsymbol{w} \succcurlyeq 0, \boldsymbol{\mu} \succcurlyeq 0, \boldsymbol{\nu} \succcurlyeq 0, \beta} \min_{\boldsymbol{\lambda}, \rho, \boldsymbol{\psi}} \mathcal{L}(\boldsymbol{\lambda}, \rho, \boldsymbol{\psi}, \boldsymbol{w}, \boldsymbol{\mu}, \boldsymbol{\nu}, \beta)$. In order to ensure that $\min_{\boldsymbol{\lambda}, \rho, \boldsymbol{\psi}} \mathcal{L} \neq -\infty$, we need

$$
\begin{cases}
\displaystyle\sum_{i=1}^{n} w_i = 1 \\
\dfrac{1}{\alpha n} - \nu_i - w_i = 0 \Rightarrow w_i \leq \dfrac{1}{\alpha n}; \ \forall i \in [n] \\
\beta - \mu_s + \displaystyle\sum_{i=1}^{n} w_i \ell_i^s = 0 \Rightarrow \langle \boldsymbol{w}, \boldsymbol{\ell}^s \rangle \geq -\beta; \ \forall s \in [t]
\end{cases}
\tag{14}
$$

Under these conditions, we have $\mathcal{L} = -\beta - \sum_{i=1}^{n} w_i = -\beta - 1$. Let $\gamma = \beta + 1$, then the dual problem becomes

$$\max_{\boldsymbol{w} \succcurlyeq 0, \gamma} \quad -\gamma$$
$$\text{s.t.} \quad \langle \boldsymbol{w}, \boldsymbol{\ell}^s \rangle \geq 1 - \gamma; \ \forall s \in [t]$$
$$\sum_{i=1}^{n} w_i = 1, \quad w_i \leq \frac{1}{\alpha n}; \ \forall i \in [n] \tag{15}$$

which is equivalent to (5).

**Connection to Original LPBoost.** The original soft-margin LPBoost formulation (Eqn. (4) and (5) in [DBST02]) is:

**Dual:**

$$\min_{\boldsymbol{w}, \gamma} \quad \gamma$$
$$\text{s.t.} \quad \sum_{i=1}^{n} w_i y_i H_{is} \leq \gamma; \ \forall s \in [t] \tag{16}$$
$$\boldsymbol{w} \in \Delta_n, \boldsymbol{w} \preccurlyeq D$$

**Primal:**

$$\max_{\boldsymbol{\lambda}, \rho, \boldsymbol{\psi}} \quad \rho - D \sum_{i=1}^{n} \psi_i$$
$$\text{s.t.} \quad \psi_i \geq \rho - y_i \langle H_i, \boldsymbol{\lambda} \rangle, \psi_i \geq 0; \ (\forall i \in [n])$$
$$\boldsymbol{\lambda} \in \Delta_t \tag{17}$$

where $H \in \mathbb{R}^{n \times t}$ is some matrix and $\boldsymbol{y} \in \mathbb{R}^n$ is some vector. Now, let $D = \frac{1}{\alpha n}$, $y_i = 1$ for all $i \in [n]$, and $H_{is} = 1 - \ell_i^s$ for all $i, s$. Then, it is easy to show that the above primal-dual problem becomes the $\alpha$-LPBoost primal-dual problem (5) and (6).

## A.4 Proof of Proposition 3

The proof is based on the following dual formulation of CVaR (see Example 3 in [DN18]):

$$\text{CVaR}_\alpha^\ell(F) = \min_{\eta \in \mathbb{R}} \left\{ \alpha^{-1} \frac{1}{n} \sum_{i=1}^n \left( \ell(F(x_i), y_i) - \eta \right)_+ + \eta \right\} \tag{18}$$

So we have

$$
\begin{aligned}
\rho_*^t &= \max_{\boldsymbol{\lambda} \in \Delta_t} \max_{\rho \in \mathbb{R}} \left( \rho - \frac{1}{\alpha n} \sum_{i=1}^n (\rho - 1 + \sum_{s=1}^t \lambda_s \ell_i^s)_+ \right) \\
&= \max_{\boldsymbol{\lambda} \in \Delta_t} - \min_{\rho \in \mathbb{R}} \left( \frac{1}{\alpha n} \sum_{i=1}^n (\rho - 1 + \sum_{s=1}^t \lambda_s \ell_i^s)_+ - \rho \right) \\
&= \max_{\boldsymbol{\lambda} \in \Delta_t} - \min_{\eta \in \mathbb{R}} \left( \frac{1}{\alpha n} \sum_{i=1}^n (\sum_{s=1}^t \lambda_s \ell_i^s - \eta)_+ - 1 + \eta \right) \qquad (\eta = 1 - \rho) \\
&= \max_{\boldsymbol{\lambda} \in \Delta_t} \left( 1 - \text{CVaR}_\alpha^{\ell_{0/1}}(F) \right)
\end{aligned} \tag{19}
$$

since $\ell_i^s$ is defined as the zero-one loss of model $f^s$ over $z_i$. And since the primal problem finds the $\boldsymbol{\lambda}^*$ that maximizes $\rho_*^t$, $\boldsymbol{\lambda}^*$ achieves the maximum above. $\qquad\square$

## A.5 Proof of Theorem 5

Consider an expert problem where there are $n$ experts such that the loss of expert $i$ at round $t$ is $1 - \ell_i^t \in [0, 1]$ (e.g. let the prediction of expert $i$ at round $t$ be $1 - \ell_i^t$, and let the loss function be $\ell(\hat{y}) = \hat{y}, \hat{y} \in [0, 1]$). A weighted average forecaster randomly samples an expert according to the weights $\boldsymbol{w}^t$ at round $t$, and its average loss is $r^t = \sum_{i=1}^n w_i^t (1 - \ell_i^t)$. Then Algorithm 2 satisfies $w_i^{t+1} \propto \exp(-\eta \sum_{s=1}^t r_i^s)$ for all $t$, so by Theorem 2.2 in [CBL06] we have

$$\frac{\log n}{\eta} + \frac{T\eta}{8} \geq \sum_{t=1}^T r^t - \min_{i \in [n]} \sum_{t=1}^T (1 - \ell_i^t) = \max_{i \in [n]} \sum_{t=1}^T \ell_i^t - \sum_{t=1}^T \sum_{j=1}^n w_j \ell_j^t \tag{20}$$

By assumption, for all $t$ we have $\sum_{j=1}^n w_j \ell_j^t \leq g$. With $\eta = \sqrt{\frac{8 \log n}{T}}$, we have

$$\max_{i \in [n]} \frac{1}{T} \sum_{t=1}^T \ell_i^t \leq g + \sqrt{\frac{\log n}{2T}} \tag{21}$$

Let $\delta = \sqrt{\frac{\log n}{2T}}$, then $T = O(\frac{\log n}{\delta^2})$. Finally, note that the $\alpha$-CVaR zero-one loss of the ensemble model is upper bounded by $\max_{i \in [n]} \frac{1}{T} \sum_{t=1}^T \ell_i^t$. $\qquad\square$

# B Experiment Details

On the COMPAS dataset, we use a three-layer feed-forward neural network activated by ReLU as the classification model. For optimization we use momentum SGD with learning rate 0.01 and momentum 0.9. The batch size is 128. We warmup the model for 3 epochs, and each base model is trained for 500 iterations, with the learning rate 10x decayed at iteration 400.

On the CelebA dataset, we use a ResNet18 as the classification model. For optimization we use momentum SGD with learning rate 0.001, momentum 0.9 and weight decay 0.001. The batch size is 400. We warmup the model for 5 epochs, and each base model is trained for 4000 iterations, with learning rate 10x decayed at iteration 2000 and 3000.

On each of the Cifar-10/Cifar-100 dataset, we take out 5000 samples from the training set and make them the validation set. The remaining 45000 training samples consist the training set. We use a

WRN-28-1 on Cifar-10 and a WRN-28-10 on Cifar-100. For optimization we use momentum SGD with learning rate 0.1, momentum 0.9 and weight decay 0.0005. The batch size is 128. For Cifar-10, we warmup the model for 20 epochs, and each base model is trained for 5000 iterations, with the learning rate 10x decayed at iteration 2000 and 4000. For Cifar-100, we warmup the model for 40 epochs, and each base model is trained for 10000 iterations, with the learning rate 10x decayed at iteration 4000 and 8000.

On all datasets and for all $\alpha$, we use $\eta = 1.0$ for $\alpha$-AdaLPBoost.