# OpenReview forum: " Boosted CVaR Classification"
_NeurIPS.cc/2021/Conference — NeurIPS 2021 Poster_

### Official Review · Reviewer_7rtp · 2021-06-26

**Rating:** 5
**Confidence:** 4

**Summary:**

In this paper, the authors proposed two boosting algorithms named the $\alpha$-LPBoost method and the $\alpha$-AdaLPBoost (an efficient model) to solve the CVaR problem under the zero-one loss condition. Through the experiments and theoretical analysis, they claim that the minimization of the CVaR problem with randomized models under the 0-1 loss condition can achieve a lower loss than using the deterministic model.

**Limitations And Societal Impact:**

This paper only considers 0-1 loss. In practice, we usually apply some surrogate losses for 0-1 loss because it is non-differentiable. The methods and results in this paper are hard to be generalized. I suggest the authors should discuss this limitation in this work.

**Main Review:**

In general, the proposed method, which modifies the CVaR problem to an LPboosting problem is new. The authors demonstrate the difference between their work and existing works. This submission is technically sound. Some claims are well supported by theoretical analysis and experiments. The paper is well-written in several sections. However, the experiment part is not clear enough and contains many typos.

My major comments are as follows,

1.  In the abstract (lines 17-18), the authors mentioned the proposed method can improve the tail performance by a large margin compared to the deterministic model. However, in the experiment part, the authors only compare the losses in different $\alpha$ conditions. The only conclusion is the proposed method has a lower loss than the compared methods. How can the authors get a high tail performance conclusion from the experiment?  There are no numerical results (i.e., accuracy) that we can refer to.

2. In the experiment part, there is missing some initial setting information about the proposed method and the compared methods. Specifically, in Figures 1 and 2, can we get the best performance from ERM or AdaBoost+Average method if we tune the hyperparameters or setting different initial weights? In Figure 3,  can the regularized $\alpha$-LPBoost method better than the $\alpha$-AdaLPBoost if we use the average performance from setting several random seeds or we set a different $\beta$ parameter? Therefore, the sensitivity experiments in this paper are missing. Furthermore, 0-1 loss is a non-differentiable loss. Without large sensitivity experiments, it is hard to judge your model performance and experiment results.

3. Some significant proofs are missing in the main paper or in the appendix. For example,  in line 131, how to derive $\alpha$-CVaR from the limit of DRO with Renyi-divergence? In line 180, how to derivative dual formulation from the primal formulation? Since your dual and primal formulations are different from LPBoost, you cannot just provide a citation in line 184.

4. In line 143, the $\alpha$-CVaR loss is bounded by 1 and ERM. According to your formulation, we can only say $\alpha$-CVaR loss is non-decreasing with the ERM loss. How can you say 'monotonically increases'?

5. Since this paper focus on the CVaR, why the authors do not design an obvious baseline method, which optimizes CVaR directly? If directly optimize CVaR achieves the best performance, why we need to use the boosting method?

My minor comments are as follows,

1. Experimental part is not well-written. There are some typos. For example, in line 266, 'we too adopt' should be 'we adopt'. In line 168 and line 269, ' Cifar' should be 'CIFAR', '100-class classification' should be '100-class', 'On COMPAS' should be 'On COMPAS, '. The authors should do several proofreadings

**Time Spent Reviewing:**

5

---

> ### Author Response · Authors · 2021-08-10
> **Response to Reviewer 7rtp**
>
> We thank the reviewer for the helpful comments. We would like to address your concerns as follows:
>
> 1. Why do experiments show improvement in in some $\alpha$-losses rather than tail performance as promised in the abstract: we note that the $\alpha$-CVAR zero one loss is precisely what we (and other related papers) mean by tail performance: it explicitly measures the accuracy on the worst $\alpha$-fraction (i.e. the tail) of the data (see lines 118-121). From Fig. 1, we can see that when $\alpha$ is small, the tail performance of a randomized model, i.e. its accuracy over the worst $\alpha$ fraction of the data, is much higher than that of a deterministic model. We will add additional clarifications to the final version.
>
> 2. Sensitivity experiments: We ran the experiments repeatedly with different random seeds and the results were very consistent. We will add error bars to the figures to demonstrate the consistency. We will also include experiments on how the hyperparameters affect the tail performance in the appendix.
>
> 3. For deriving CVaR from the limit of DRO with Renyi-divergence, please refer to Example 3 in [DN18]. We will add a derivation of the primal/dual form of $\alpha$-LPBoost in the appendix.
>
> 4. The CVaR loss is non-decreasing with the ERM loss but not monotonically increasing. We will correct this in the new version.
>
> 5. Regarding comparing with directly optimizing CVaR: We have proved in this paper that
>
>    - Optimizing the CVaR loss or any other algorithms cannot work better than ERM if we use deterministic models (Prop. 1 and l. 144-145).
>
>    - Maximizing the $\alpha$-LPBoost objective is equivalent to directly minimizing the CVaR loss for ensemble models (Prop. 3 and l. 188-189). And we compared regularized $\alpha$-LPBoost with $\alpha$-AdaLPBoost (our main algorithm) in the experiments (Figure 3).
>
>     Therefore, we believe that we have had a sufficient discussion on this point.
>
> 6. Regarding not using surrogate losses: This is a major misunderstanding and we would like to clarify. Our framework consists of two parts: (i) an unfair learner that produces models with low 0/1 loss which we assume we have access to (l. 194-196); (ii) the Boosting part that picks sample weights for this unfair learner. In practice, the unfair learner is usually implemented by minimizing a surrogate loss (we implement the unfair learner by minimizing the cross-entropy loss in our experiments, so we do use a surrogate loss). On the other hand, the Boosting part only picks sample weights and does not require any gradients, so a differentiable surrogate loss is not necessary. We use the 0/1 loss for the Boosting part instead of a surrogate loss because the classifier is evaluated by the 0/1 loss (or accuracy), so using the 0/1 loss is more direct. Does the use of 0/1 loss cause computational intractability here? No! We just need to solve a simple LP (Algorithm 1), or a closed form expression (Algorithm 2) to get the sample weights. The original AdaBoost also uses the 0/1 loss (i.e. whether each sample is correctly classified) for picking sample weights. So overall, our approach does not rely on any computationally hard problems such as minimizing zero-one loss: Algorithms 1 and 2 can be seen to not involve any intractable step! Moreover, the results can be generalized by implementing the unfair learner in different ways.
>
> We will do several proofreadings and correct the typos in the current version. We hope that our response answers your questions.
>
> [DN18] John Duchi and Hongseok Namkoong. Learning models with uniform performance via distributionally robust optimization. arXiv preprint arXiv:1810.08750, 2018.

---

> > ### Comment · Reviewer_7rtp · 2021-08-17
> > **Keep my rating**
> >
> > Hi Authors,
> >
> > Thank you so much for your response! I have some comments about your response.
> >
> > 1. You mentioned 'we note that the $\alpha$-CVAR zero one loss is precisely what we (and other related papers) mean by tail performance'. I agree with this. However, you mentioned 'it explicitly measures the accuracy on the worst -fraction (i.e. the tail) of the data (see lines 118-121)'. I do not agree with this one. CVaR quantifies the expected losses that occur beyond the VaR breakpoint. Why it is the accuracy? Your definition of' tail performance is the accuracy over the minority groups' (line 24) is not precise. Please let me know if you can find any materials that can support your statement. On the other hand, your Eq.(2) is the loss instead of accuracy. If you would like to learn more about CVaR. I suggest you should read Section 6 of the book [1]. As I mentioned before, there are no numerical results (i.e., accuracy) that we can refer to. Your experiments only evaluate the loss.
> >
> > 2. As I mentioned, the sensitivity experiments in this paper are missing, which you have confirmed.
> >
> > 3. As I mentioned, how to derivative dual formulation from the primal formulation is missing, which you have confirmed.
> >
> > 4. As I mentioned, your 'monotonically increases' statement about CVaR is wrong, which you have confirmed.
> >
> > 5. I agree you do not need to consider the non-differentiable problems. However, your paper only focuses on the 0-1 loss. So I do not know the applicability of your algorithm in practice.  And you should mention this limitation in this paper.
> >
> > Therefore, I intend to keep my score for this paper.
> >
> > [1] Shapiro, Alexander, Darinka Dentcheva, and Andrzej Ruszczyński. Lectures on stochastic programming: modeling and theory. Society for Industrial and Applied Mathematics, 2014.

---

> > > ### Author Response · Authors · 2021-08-22
> > > **Response to Your New Questions**
> > >
> > > Thank you so much for your reply. We would like to address your further concerns as follows:
> > >
> > > 1. Regarding why the CVaR 0/1 loss measures classification accuracy: In machine learning, we use the 0/1 loss to measure the accuracy of a classifier because for any classifier, its average 0/1 loss = 1 - its accuracy. The CVaR$_{\alpha}$ loss which we define in Eqn. (2) measures the loss of the classifier over the worst $\alpha$ fraction of the data, and when that loss is chosen as the 0/1 loss, the CVaR 0/1 loss = 1 - the accuracy over the worst $\alpha$ fraction of the data. We never said that the CVaR loss can measure the accuracy. What we said is that the "CVaR 0/1 loss" and the "Average 0/1 loss" measure the tail and overall classification accuracy. In all figures in the experiments section we report the "CVaR 0/1 loss" or the "Average 0/1 loss", which do measure the accuracy.
> > >
> > > 2. Regarding the definition of tail performance: In this paper, we define the tail performance of a classifier as "its accuracy over the worst fraction of the data". The definition in line 24 is widely used "in the field of algorithmic fairness" (line 23) (See e.g. [HSNL18]) and is not used in this paper. We briefly talk about this definition for the sole reason of introducing related areas.
> > >
> > > 3. Regarding not using surrogate losses: We would like to point out that our method can be extended to any other loss (we can boost or upweight the samples with higher losses). Nevertheless, we do not see any point in using losses other than the 0/1 loss within the context of this paper since we solely focus on classification tasks and the 0/1 loss directly evaluates the accuracy of a classifier. In practice a classifier is evaluated by its accuracy (which is "1 - its average 0/1 loss"), and we only use differentiable surrogate losses for applying gradient methods during training, which as we mentioned is not required for Boosting.
> > >
> > > We hope that our response could address your new concerns.
> > >
> > > [HSNL18] Tatsunori Hashimoto, Megha Srivastava, Hongseok Namkoong, and Percy Liang. Fairness without demographics in repeated loss minimization. ICML 2018.

---

### Official Review · Reviewer_aDDc · 2021-07-18

**Rating:** 5
**Confidence:** 3

**Summary:**

The paper attacks the problem of "fair classification", where the goal is to train a classifier that is "good" on any small fraction of the samples. Obviously, any deterministic classifier having less than 100% accuracy is arbitrarily bad on a small fraction of the distribution. The authors propose the usage of randomised classifiers to bypass this. i.e we prefer a classifier that is correct on all samples 80% of the time over a classifier that is correct on 80% of the samples wrong on the rest 20% all the time.

Firstly the authors observe the superiority of randomised classifiers for the CVaR loss over deterministic classifiers.  Then they propse the usage of a boosting framework to aggregate individual "good but unfair" classifiers into "equally good but fair" classifiers. In particular, the classic LP-boost algorithm is observed to be a good fit for this problem.

The authors then propose a more efficient version of the LP-boost algorithm where classifiers need not be trained for different levels of fairness (the worst fraction of samples being optimised for).

The algorithms are validated on classic classification datasets where they do better than standard baselines.

**Limitations And Societal Impact:**

Yes.

**Main Review:**

The observation of CVaR for randomized classifiers being better is interesting but not surprising or significant.

The equivalence of LP-Boost to alpha CVaR is very interesting while still being obvious in hindsight and hence deserves credit.

The alpha-LP boost algorithm for CVaR along with AdaLP boost are modestly significant.


Some other comments:

1. The usage of randomised classifiers for fairness has technical issues. A classifier often needs to perform in the same way when applied repeatedly. E.g. Otherwise in a loan application classifier, the individual can just apply repeatedly until he gets through.

2. Rename the function ERM[f] to zero-one[f] or similar. (line 113)

3. Make the defintion of CvAR valid for randomised classifiers too, (line 119)

4. Give a better example in lines 150 to 156, such that a CVaR  of less than 0.5 can be achieved. As a simple coin toss can achieve 0.5

5. How is eta chosen in AdaBoost+average?

6. If eta is fixed there is no Adaptiveness in "adaBoost", and hence a better name might be boost by majority.

7. By the theorems the the CvAR loss should l=never go above g, the guaranteed performance of the unfair learner. The NN learners surely contain the constant function which would predict all classes with equal probability. So why does AdaBoost+average and AdaLPboost have losses greater than 0.5 in figure 1c, and loss greater than 0.9 in Figure 1b for alpha<0.1?  0.5 and 0.9 are the performance of a random classifier for 2-class and 10-class problem respectively.

8. Are there standard bounds linking the train CVaR loss to the test loss, similar to the margin bounds for standard zero-one loss? If so, references to these would be helpful.





**Time Spent Reviewing:**

6

---

> ### Author Response · Authors · 2021-08-10
> **Response to Reviewer aDDc**
>
> We thank the reviewer for the helpful comments. We would like to address your  concerns as follows:
>
> 1. Regarding the significance of this work: Although the results may not surprising in hindsight, our work makes three significant contributions:
>
>    - This work is a response to the pessimistic result in [HNSS18] that no algorithm can achieve higher tail performance than ERM for classification. It provides a simple solution to an open problem.
>
>    - This work proposes a boosting framework for fairness, which we believe is novel and interesting. Our results show that an unfair learner can be boosted to a fair one.
>
>    - We show that there is a direct connection between training fair models (CVaR) and Boosting (LPBoost). Future works can rely on this connection to design better algorithms for fairness.
>
>
> 2. Can someone game randomized classifiers via repeatedly using them e.g. for credit decisions: we note that we actually provide high probability guarantees rather than only in expectation (see e.g. Eqn. 9). So the user would need at least O(1/delta) uses to see bad tail performance, which might not always be feasible. Let's contrast this with deterministic classifiers: we have explicitly shown (see Prop. 1) that they are no better than ERM, so they are guaranteed to have bad tail performance! Overall, game-ability is an interesting general research question in the field of decision making, though out of scope of the paper, we will definitely add the interesting discussion above to the paper.
>
> 3. Regarding comments 2-4: We thank the reviewer for these very good comments and will edit the paper as suggested.
>
> 4. Regarding $\eta$: In Thm.5, $\eta = \sqrt{8 \log n /T }$ (Appendix A.4). In our experiments, for all datasets we choose $\eta = 1.0$ which is close to this theoretical value (Appendix B).
>
> 5. Regarding the name "AdaBoost": This is a minor point, but "AdaBoost is adaptive in the sense that subsequent weak learners are tweaked in favor of those instances misclassified by previous classifiers." (from Wikipedia) We believe that our method has this property, so it can be called "adaptive".
>
> 6. The reason why sometimes the tail performance is lower than the theoretical value in the experiments is that the number of base models $T$ we use is not large enough. In the proof of Thm. 5 we show that we need $T \ge \log n / 2\delta^2$. e.g. For CIFAR-10, $n=50000$, so if we want to achieve $\delta=0.1$, then we need $T > 500$. Using such a large $T$ will take an unreasonable amount of time. Figure 4 also shows that as $T$ increases, the tail performance becomes better but the convergence is slow. We will add a remark on this issue in the experiments section.
>
> 7. Regarding generalization bounds for CVaR: In a recent work, [KLPR20] derive uniform convergence bounds for CVaR loss. These results can be used in a straightforward manner to derive Rademacher complexity based generalization bounds of our classifier. We will add this result to our paper.
>
> We hope that our response answers your questions.
>
> [HNSS18] Weihua Hu, Gang Niu, Issei Sato, and Masashi Sugiyama. Does distributionally robust supervised learning give robust classifiers? ICML 2018.
>
> [KLPR20] Justin Khim, Liu Leqi, Adarsh Prasad, and Pradeep Ravikumar. Uniform convergence of rank-weighted learning. ICML 2020.

---

### Official Review · Reviewer_h6bZ · 2021-07-19

**Rating:** 8
**Confidence:** 5

**Summary:**

This paper studies the boosting framework for classification. Weak classifiers are produced sequentially by an abstract learner. The popular α-CVaR (Conditional Value at Risk) is adopted. The 0/1 loss is considered. First, it is proved that the optimizer (among all the deterministic models) of ERM is also the optimizer of α-CVaR, and vice versa if α-CVaR could be smaller than α by some deterministic model. The authors also prove that with the same 0/1 loss, the randomized model achieves smaller α-CVaR. The optimization structures of α-LPBoost and α-AdaLPBoost are studied to construct randomized classifiers.

**Ethical Concerns:**

N.A.

**Limitations And Societal Impact:**

Limitations: see our concerns above

Societal impact: big potential of societal impact through the significant improvement of classification algorithm for machine learning and data science.

**Main Review:**

Main concerns:

1. Evidently the generalization ability of boosting depends on the richness of weak classifiers. The link between learner and generalization power is not discussed. In particular, we believe that with the 0/1 loss, such a learner would not be computationally tractable.
2. How does the proposed Boosted CVaR Classification framework learn randomized classifier? We note that in remark 2 at the end of page 4, F is defined as a function; however, F is defined as a tuple before Equation (4), which is unlikely to produce a number.

some points:
1. Line 131, "DRO" should be "DRO^\ell"
2. We are not certain about the claim "... then the α-CVAR loss is equal to the limit of ..." in lines 130--131. Apparently, the α-CVAR loss is defined in (2) based on n sample points and the unit simplex of R^n, but the limit discussed here has no relation with n.
3. What is the definition of "deterministic model"? In particular, is it more than a map in the general sense? If so, is there any example of a map that is a model but not deterministic? Moreover, what is a "randomized model"? We note footmark 2 on page 4 on the randomized model, but that is a description of its property, and it is not a rigorous definition.
4. In the proof of Proposition 2, why is the CVaR_alpha loss of F' no greater than 1?

==================================

After reading the Author's/Authors' response, I think that all of my technical concerns have well been addressed. I appreciate the careful and comprehensive response by the Author/Authors. I would like to raise my rating.

**Time Spent Reviewing:**

5 hours

---

> ### Author Response · Authors · 2021-08-10
> **Response to Reviewer h6bZ**
>
> We thank the reviewer for the comments and would like to address the concerns as follows:
>
> 1. Regarding the tractability of the learner: We strongly disagree with the claim that "with the 0/1 loss, such a learner would not be computationally tractable". We would like to remind the reviewer that the main assumption of our framework is access to an unfair learner that produces base classifiers whose 0/1 loss is lower than a constant $g$ (as clearly stated in l. 194-196). We don't need the unfair learner to exactly minimize the 0/1 loss. Such learners are easy to implement, e.g. a learner which minimizes the cross entropy loss of the classifier achieves low 0/1 loss in a number of settings. Provided with such an unfair learner, Boosting picks the sample weights for this learner and does not require any gradients, so a differentiable loss is not required for the Boosting part. Both the classic AdaBoost (which boosts a weak learner to a strong one) and most of its variants, and the Boosting we propose (which boosts an unfair learner to a fair one) work with 0/1 loss, and we also verify our method with experiments on real datasets, so we cannot see why such a learner could not be computationally tractable.
>
> 2. Regarding the generalization ability of our technique: Previous works showed that Boosting has good generalization guarantees (see e.g. [GKLMN19] and its references). The results in these works entail that the classifiers learned using our technique generalize well. In future versions of the paper, we will add a concrete theorem on generalization guarantees of our technique.
>
> 3. The notation $F=(f^1,\cdots,f^T,\lambda)$ means that the ensemble model $F$, which is a function, is composed of the base classifiers $f^1,\cdots,f^T$ and the model weight vector $\lambda$. How the ensemble model works at inference time is described in l. 165-168. We will make the notation clearer in the new version.
>
> 4. The $\alpha$-CVaR loss is equal to the limit of the DRO loss with Renyi divergence "as $\beta \rightarrow \infty$" (l. 131), not "as $n \rightarrow \infty$". Please refer to Example 3 in [DN18] for the proof of this statement.
>
> 5. A deterministic model is a deterministic mapping $F: X \rightarrow Y$ (l. 134), while the output of a randomized model is a random variable, such that $P(F(x) = y)$ for any pair $(x,y)$ is in $[0,1]$ instead of binary (Footnote 2). We will make the definition clearer in the new version.
>
> 6. The CVaR$^{\ell_{0/1}}_\alpha$ loss of $F'$ is no greater than 1, because it is a maximum expectation of the 0/1 loss (Eqn. (2)).
>
> We hope that our response answers your questions.
>
> [GKLMN19] Allan Grønlund, Lior Kamma, Kasper Green Larsen, Alexander Mathiasen, and Jelani Nelson. Margin-based generalization lower bounds for boosted classifiers. NeruIPS 2019.
>
> [DN18] John Duchi and Hongseok Namkoong. Learning models with uniform performance via distributionally robust optimization. arXiv preprint arXiv:1810.08750, 2018.

---

### Official Review · Reviewer_6kwQ · 2021-07-21

**Rating:** 6
**Confidence:** 3

**Summary:**

The paper first shows that for deterministic models, optimizing for tail performance (in terms of 0/1 loss) is in fact equivalent to optimizing for average performance, and thus doesn't actually improve fairness of a model. As a mitigation, it is proposed to switch to non-deterministic classifiers, for which this property no longer holds. A boosting algorithm that can construct such a $\alpha$-CVaR optimized non-deterministic classifier based on training weighted deterministic unfair classifiers is proposed and demonstrated to provide good results.

**Limitations And Societal Impact:**

I think one limitation is that without knowing anything about the subgroups, it is impossible to determine whether a data point comes from an underrepresented subgroup, or is just a mistake in the ground-truth labeling.

Also, for a practical application towards fairness, some way of determining the value for $\alpha$ would be needed I think.

**Main Review:**

The first part of the paper is to motivate the use of non-deterministic classifiers (or equivalently a scorer that outputs probabilities) because on deterministic classifiers any ERM minimizer also minimizes the CVaR and as such minimizing for CVaR would not actually get fairer classifier. This is a special case that follows from a known more general result, but the paper provides a simple direct proof for the statement. It is then shown that when going to non-deterministic classifiers, 1) this equivalence no longer holds [Prop 2] and 2) the classifier can be learned using a variation of LP boosting [Prop 3], which is demonstrated to be effective. The ability to train classifiers that perform well on minority subgroups is an important problem nowadays, so I think that the contribution of the paper is significant.

Below is a list of things I noticed while reading, and some questions that I had:

# Details:
* l. 137 maybe explicity put the words "it holds that" or similar into the sentence. Right now, it is only clear that the CVaR statements are the claim once you've reached the period at the end of the sentence.
* Section 3.1 I think a bit more detail would be helpful here, i.e. an explanation where the primal/dual problems come from -- what are they trying to optimize. The formulas look a bit different from LPBoost,
I guess because of the switch from {-1, 1} to [0, 1]. Any intuition, besides that this is what is needed to match the expression against CVar in the proof of Prop 3?
* l. 182 The way the problem is formulated here is technically nonlinear, since its objective contains the max function
* l. 187 I think the statement as it is written here is not as strong as it should, as (7) just states that there exists a $\lambda$ such that the optimal value coincides with 1 minus minimum CVar.
However, the proof in the appendix seems to show that the lambdas are actually the same, I think, so that works out in the end and your statement in line 198 is actually true.
* l. 278 How did you train to minimize the empirical risk? Minimizing categorical cross entropy? If so, are the ERM values in Figure 1/2 corresponding to a thresholded model, or to the probabilities learned by cross-entropy?
* Figure 1/2 As the theory is based on the losses on training data, it would be nice to have (at least in the appendix) these graphs also for training losses.
* l. 292: Should be "minimum" here I think
* Appendix: In the appendix, it would be nice if the proofs were given together with the full assumptions and claim.

* A1:Given that this is in the appendix and thus space is not such a pressing concern, maybe add one or two intermediate steps. For example,
the min in line 4 appears to come from whether there are more than $\alpha n$ misclassified examples (then we can just put all weight on them)
or less (then we can only put $1/(\alpha n)$ on each). If you define $M$ as the set of misclassified examples, then the CVaR is either
$(1/|M|) |M| = 1$ or $1/(\alpha n) |M|$.
* l. 6 I don't get the implication here? If the minimum is less than one, than clearly there exists an $F$ for which the statement is true, but why is it true for all $F$?

=====
Update

Taking into account the author's comments, I have raised my score to slightly above.
Based purely on the content, I might have given a higher rating, but I think that the clarity of the writing in the paper should be improved more before qualifying as "Good paper, accept".

**Time Spent Reviewing:**

6

---

> ### Author Response · Authors · 2021-08-10
> **Response to Reviewer 6kwQ**
>
> We thank the reviewer for the detailed and helpful comments. We would like to address your concerns as follows:
>
> 1. We will rephrase the sentence in line. 137 as suggested.
>
> 2. $\alpha$-LPBoost is a variant of LPBoost, and we use this variant because its objective has a direct connection with the CVaR objective (Prop. 3). We will add some intuitions on the formulation of $\alpha$-LPBoost in Section 3.1, and a comparison between $\alpha$-LPBoost and the original LPBoost in the appendix.
>
> 3. The $\alpha$-LPBoost is a linear program. Though the primal form contains a max, it can be rewritten as a linear program by introducing slack variables $\psi_i$: maximize $\rho - (\alpha n)^{-1} \sum_i \psi_i$ such that $\psi_i \ge 0$, $\psi_i \ge \rho - 1 + \sum_s \lambda^s \ell^s_i$, $\lambda \in \Delta_t$. We will add a derivation of this formulation in the appendix.
>
> 4. Regarding the same $\lambda$: This is a very good point and we thank the reviewer for pointing it out. We will add this point to Prop. 3.
>
> 5. We train the ERM models by minimizing the cross entropy loss. The ERM values in Figures 1/2 correspond to thresholded models.
>
> 6. We will add the plots of the training losses in the appendix.
>
> 7. l. 292: You are right. We will correct this typo.
>
> 8. Regarding the proof of Prop. 1: The implication in l. 6 is true because given $ CVaR_{\alpha}^{\ell_{0/1}}(F) = min \lbrace 1, \alpha^{-1} ERM^{\ell_{0/1}}(F) \rbrace $, if $\text{ERM}^{\ell_{0/1}}(F) < \alpha$ for all $F$ (as the minimum is less than $\alpha$), then $CVaR_{\alpha}^{\ell_{0/1}}(F) = \alpha^{-1} ERM^{\ell_{0/1}}(F)$ for all $F$. We will add full assumptions and claims and more intermediate steps to make the proofs easier to understand.
>
> 9. Labeling mistakes and outliers in the dataset can have a negative effect on fair training algorithms. A very recent paper [ZDKR21] studied this issue and proposed a solution called DORO, which can be combined with the method we propose. We will add a brief remark on this issue, but it is not the main focus of this paper.
>
> 10. Finding some way to determine the value of $\alpha$ is an interesting open question which to our best knowledge has no general solutions yet, and we will explore this question in the future. We would like to emphasize that the problem we study in this paper is how to solve the computational problem of maximizing the performance over the worst $\alpha$ fraction of the data for a given $\alpha$ (l. 29-42). Determining the value of $\alpha$ is not required for the problem we study.
>
>
> We hope that our response answers your questions.
>
> [ZDKR21] Runtian Zhai, Chen Dan, Zico Kolter, and Pradeep Ravikumar. Doro: Distributional and outlier robust optimization. ICML 2021.

---

> > ### Comment · Reviewer_6kwQ · 2021-08-29
> > **Proof of Thm 5**
> >
> > Sorry for the late response.
> > I wanted to update and finalize my review and finally found the time to check Theorem 5, but I'm having trouble following the proof, could you maybe elaborate?
> >
> > The proof seems to rest on an application of Thm 2.2 of [CBL06], which is a regret bound for weighted average predictors constructed from a set of experts.
> > In the setting here, if I understand things correctly, the experts are deterministic classifiers that have been trained in some way, and as such their output space is a discrete set, contrary to CBL06 which assume a convex subset of a vector space. I guess you assume this to be embedded into the $[0, 1]^d$ hypercube and do a linear extension of the 0-1 loss to this subset. Because of the linearity, also 1 - l is a linear (thus convex) function, an therefor Thm 2.2 is applicable? But this argument breaks if you replace 0-1 with alpha-CVaR-0-1 loss I think?
> >
> > Also, conceptually the structure of the proof seems a bit weird to me. The crucial argument in the end appears to be that the loss of the weighted average cannot be worse than that of the worst expert model (basically turning around the typical regret bounds using the 1- l loss formulation). The rest of the proof is just figuring out an upper bound for the worst expert. Apparently, this upper bound can be found using weighted average classifier, even though it is completely independent of the averaging. Do you have any intuition for that?
> >
> > Finally, you are using "by assumption" that for all t
> > $$\sum_{i=0}^n w_j l_j^t \leq g,$$
> > but the corresponding assumption in the main text has $n$ as the number of samples, whereas here $n$ sums over the experts. Maybe I'm missing something fundamental about the setup here?

---

> > > ### Author Response · Authors · 2021-08-29
> > > **Elaboration on the Proof of Theorem 5**
> > >
> > > Thank you so much for the reply. We would like to elaborate on the proof of Thm. 5:
> > >
> > > 1. The experts are not "deterministic classifiers that have been trained in some way". We would like to first clarify what an expert problem is:
> > >     - There are $n$ experts.
> > > 	- At the beginning of each round, we need to choose one expert and follow its prediction in this round.
> > > 	- After we make a choice, each expert makes its prediction and incurs some loss which is then revealed to us.
> > > 	- Thm 2.2 in [CBL06] says that if we use an exponential weighted average forecaster, then the expected loss we incur (which is accumulated loss of all expert predictions we chose) is very close to the loss of the "best" expert (with the lowest accumulated loss) (not "worst"!).
> > >
> > >     Note that we do not actually need to train the experts. What we only need to do is to define the loss function and the $n$ experts (i.e. what each expert predicts in each round).
> > >
> > >     Returning back to the proof, what we are doing here is simply constructing an expert problem with $n$ experts without training any expert. You can think of the loss function as $\ell(\hat{y},y) = \hat{y} (\hat{y} \in [0, 1])$, which is convex in $\hat{y}$ and falls in $[0,1]$. In round $t$, the prediction of expert $i$ is $1 - \ell_i^t$, so that its loss is also $1 - \ell_i^t$. Our weighted average forecaster chooses expert $i$ with probability $w_i^t$, so our expected loss in round $t$ is $\sum_{i} w_i^t (1 - \ell_i^t) = r^t$.
> > >
> > >     After $T$ rounds, our expected accumulated loss would be $\sum_{t} r^t$, while the accumulated loss of the best expert would be $\min_{i} \sum_{t} (1 - \ell_i^t)$. By Thm 2.2 in [CBL06], these two are close, which is Eqn. (13).
> > >
> > > 2. In the proof, we actually proved that the 0/1 loss of the ensemble model on the worst single sample is at most $g + \delta$ (Eqn. (14)). The $CVaR_\alpha$ 0/1 loss is the average 0/1 loss over a bunch of samples (the worst $\alpha$ fraction of the samples), so the CVaR$_{\alpha}$ 0/1 loss is upper bounded by the 0/1 loss over the worst sample, which is upper bounded by $g + \delta$. This answers the last question in your second paragraph.
> > >
> > > 3. As we mentioned, we are comparing the loss of the weighted average forecaster with the loss of the "best" expert, not the "worst" expert.
> > >
> > > 4. $n$ is the number of samples in the training set. And in our expert formulation, we construct $n$ experts, with each expert corresponding to one sample, so $n$ is also the number of experts. The assumption you asked in your third point can be found in line 208. There is a small typo in the proof: $w_j$ should be replaced by $w_j^t$.
> > >
> > > We hope that our response could help you understand the proof of Thm. 5.

---

> > > > ### Comment · Reviewer_6kwQ · 2021-08-30
> > > > **Proof of Thm 5**
> > > >
> > > > Thanks, I think that helped quite a bit :)

---

### Official Review · Reviewer_xo9P · 2021-07-25

**Rating:** 7
**Confidence:** 3

**Summary:**

The authors of this paper propose robust boosting algorithms for minimzing the "CVaR_{alpha} 0/1-loss", defined as the worse case 0/1 loss over all alpha-sized sub-populations of the entire population. Models with low "CVaR_{alpha} 0/1-loss" help ensure fairness by not performing too poorly on any individual subpopulation.

The authors provide a number of contributions:

1. They show that while optimizing "CVaR_{alpha} 0/1-loss" is equivalent to minimizing average 0/1 loss when one is restricted to using deterministic classifiers, this is not the case if one is allowed to use randomized classifiers. And in fact, randomized classifiers can in principle always perform at least as well on CVaR_{alpha} 0/1 loss as deterministic ones.

2. They use the observation in #1 to motivate boosting algorithms that explicitly optimize the CVaR_{alpha} 0/1 loss. Here they provide 4 related learning algorithms
  * alpha-LPBoost
  * Regularized alpha-LPBoost
  * alpha-AdaLPBoost
  * AdaBoost + average

3. For the proposed algorithms, the authors give complexity guarantees that bound the ensemble size as a function of the required accuracy.

4. The authors compare their CVaR-optimizing, robust boosting algorithms to the baselines of stanard boosting with empirical risk minimization (ERM and "AdaBoost + average". Their experiments show that alpha-AdaLPBoost performs well on CVaR_{alpha} 0/1 loss across all values of alpha, in contrast to ERM which performs well when alpha is large and alpha-AdaLPBoost which performs well when alpha is small.



**Limitations And Societal Impact:**

The authors address the societal limitations and potential of this work, and it is a pro-social contribution to begin with.


**Main Review:**

This paper's main contribution is to show that randomized classifiers perform better than deterministic ones on CVaR_{alpha} 0/1 loss. And the authors capitialize on this observation by developing boosting algorithms which learn mixed strategies (randomized classifiers) for this loss function, and which have guarantees.

Novelty and Originality: The finding that randomized classifiers outperform deterministic ones on CVaR_{alpha} 0/1 loss is a novel and solid finding, and it allows the authors to derive a new algorithm which directly optimizes CVaR_{alpha} 0/1 loss. So this paper provides a solid step forward by developing fair classifiers that are guaranteed not to peform too poorly on any sub-population. That said, the observation that randomized classifiers can do better in this setting than deterministic ones seems like something that would be apparent to researchers familiar with game theory and certain areas of learning theory. And the algorithms developed are built by combining and adapting previously existing boosting algorithms.


Clarity, Quality, Technical Soundness: This paper is clearly written and its claims are supported by theoretical results, as well as experimental results.

Regarding the experiments, the authors cite previous work on optimizing CVaR_{alpha} loss which they do not compare against in their experiments. For instance, the paper cites work by Hashimoto et al. 2018 on optimizing the related "DRO" loss for handling subpopulation shift. However, the paper does state that that CVaR_alpha 0/1 loss is a certain limit of the "DRO" loss. So it's not clear whether adding these comparisons would yield further insights.


**Time Spent Reviewing:**

5 hours

---

> ### Author Response · Authors · 2021-08-10
> **Response to Reviewer xo9P**
>
> We thank the reviewer for helpful comments. We would like to address the main concern raised about no comparison with previous works. We did not compare with the experimental results in e.g. (Hashimoto et al., 2018) for the following two reasons:
>
> 1. In this paper, we study the domain-oblivious sub-population shift problem where the goal is to maximize the tail performance of the model. In contrast, as discussed in lines  336-345, most previous works studied the fairness without demographics problem, where the dataset is divided into several groups and the goal is to maximize the worst-group performance of the model without knowing the group labels at training. Since the settings and objectives of the two problems are different, it is very hard to make a fair comparison. For instance, the latter usually requires some prior knowledge on the groups (e.g. a small validation set with group labels, or the groups are divided by a linear model), while we make no assumptions on the groups.
>
> 2. One might ask why not compare the tail performance between our method and previous methods like the one proposed in Hashimoto et al., 2018. The reason is that to the best of our knowledge, all previous methods use deterministic models, and we have already showed that no algorithm can achieve better tail performance than ERM if a deterministic model is used (Proposition. 1). Given that we have already provided experimental results for ERM, such a comparison wouldn't have provided additional insights.
>
> We hope that our response addresses your concern.

---

> > ### Comment · Reviewer_xo9P · 2021-08-16
> > **Acknowledgement that I have read the authors' response**
> >
> > Thank you authors. I read your response and your decisions about which methods you chose (not) to compare against. They make good sense. I will stick with my overall review.

---

### Decision · Program_Chairs · 2021-09-27

**Decision:**

Accept (Poster)

**Comment:**

The paper showed in Proposition 1 that optimizing CVaR 0/1-lossis equivalent to minimizing average 0/1 loss when one is restricted to using deterministic classifiers which motivates the usage of randomized classifiers. Although the observation is interesting but not entirely new in light of the related work [HNSS18] and other ones in learning theory, the proof given here is simpler and easier to follow.  Motivated by this, the authors proposed an efficient LP boost algorithm for optimizing CVaR with theoretical convergence analysis.

Overall, the paper presented an interesting framework based on LP-boost algorithms to build more fair classifiers with theoretical supports, although the key observation on the superior performance of randomized classifiers over deterministic is not surprising.  The revised version should take into account sensitivity analysis of hyper-parameters, the precise derivation of dual formulation, and the detailed discussion of deriving generalization bounds, and other comments made by the reviewers.